# The Impact of World Trade Center Related Medical Conditions on the Severity of COVID-19 Disease and Its Long-Term Sequelae

**DOI:** 10.3390/ijerph19126963

**Published:** 2022-06-07

**Authors:** Elizabeth Lhuillier, Yuan Yang, Olga Morozova, Sean A. P. Clouston, Xiaohua Yang, Monika A. Waszczuk, Melissa A. Carr, Benjamin J. Luft

**Affiliations:** 1Department of Medicine, Renaissance School of Medicine, Stony Brook University (SUNY), Stony Brook, NY 11794, USA; elizabeth.lhuillier@stonybrookmedicine.edu (E.L.); xiaohua.yang@stonybrookmedicine.edu (X.Y.); 2World Trade Center Health and Wellness Program, Stony Brook University (SUNY), Stony Brook, NY 11794, USA; melissa.carr@stonybrookmedicine.edu; 3Program in Public Health, Stony Brook University (SUNY), Stony Brook, NY 11794, USA; yuan.yang2@stonybrookmedicine.edu (Y.Y.); olga.morozova@stonybrookmedicine.edu (O.M.); sean.clouston@stonybrookmedicine.edu (S.A.P.C.); 4Department of Family, Population and Preventive Medicine, Renaissance School of Medicine, Stony Brook University (SUNY), Stony Brook, NY 11794, USA; 5Department of Psychology, Rosalind Franklin University of Medicine and Science, North Chicago, IL 60064, USA; monika.waszczuk@rosalindfranklin.edu

**Keywords:** 9/11 disaster, COVID-19, SARS-CoV-2, severity, comorbidities, post-acute COVID-19 syndrome

## Abstract

The individuals who served our country in the aftermath of the attacks on the World Trade Center (WTC) following the attacks of 11 September 2001 have, since then, been diagnosed with a number of conditions as a result of their exposures. In the present study, we sought to determine whether these conditions were risk factors for increased COVID-19 disease severity within a cohort of N = 1280 WTC responders with complete information on health outcomes prior to and following COVID-19 infection. We collected data on responders diagnosed with COVID-19, or had evidence of receiving positive SARS-CoV-2 polymerase chain reaction or antigen testing, or were asymptomatic but had IgG positive antibody testing. The presence of post-acute COVID-19 sequelae was measured using self-reported symptom severity scales. Analyses revealed that COVID-19 severity was associated with age, Black race, obstructive airway disease (OAD), as well as with worse self-reported depressive symptoms. Similarly, post-acute COVID-19 sequelae was associated with initial analysis for COVID-19 severity, upper respiratory disease (URD), gastroesophageal reflux disease (GERD), OAD, heart disease, and higher depressive symptoms. We conclude that increased COVID-19 illness severity and the presence of post-acute COVID-19 sequelae may be more common in WTC responders with chronic diseases than in those responders without chronic disease processes resulting from exposures at the WTC disaster.

## 1. Introduction

As of 7 March 2022, the global death toll of COVID-19, a disease caused by the SARS-CoV-2 virus, has surpassed six million [1]. Why some individuals have more severe cases of COVID-19 than others is not fully understood. A clear answer to this question has been sought since 2020, when COVID-19 first appeared, and its transmission reached a pandemic level. The initial illnesses were characterized primarily by respiratory symptoms [2], similar to other known severe coronaviruses such as severe acute respiratory syndrome coronavirus [3] and Middle Eastern respiratory syndrome coronavirus [4]. COVID-19 is now understood not only to affect the respiratory system but also to invade other crucial organ systems in the body, through the ability of the virus to bind angiotensin-converting enzyme 2, thus causing a multitude of potentially severe symptoms throughout the body [5]. Therefore, COVID-19 tends to cause severe disease not only in the lungs but also in other organ systems. 

Older age has been recognized as a major risk factor for severe COVID-19 and has been found to increase the risk of mortality [6,7,8,9,10]. Researchers have also examined various comorbidities such as obesity, chronic obstructive pulmonary disease (COPD), hypertension, diabetes, and malignant tumors as possible risk factors for severe COVID-19 and poorer health outcomes [9,11,12]. One study indicated that early-onset diabetes increases the risk of hospitalization, and obesity increases the risk of intubation and admission to an intensive care unit [13]. The U.S. Centers for Disease Control and Prevention (CDC) has emphasized as well that certain medical conditions such as cancer, obesity, chronic lung diseases, neurological conditions, and diabetes can place people at a higher risk of severe COVID-19 [14]. Asthma is another comorbidity of high interest, given that chronic respiratory diseases such as COPD are a risk factor. Yet, studies have also suggested that the presence of psychiatric conditions such as major depression may increase the risk of COVID-19 mortality in hospitalized individuals [15]. The CDC has stated that people with COVID-19 illness and moderate-to-severe asthma are at a higher risk of hospitalization [16]. We noted that in our cohort, participants had a high prevalence of conditions that predispose patients to significant COVID-19-related severity. For instance, 60% have URD, 35% have OAD, 49% have GERD, and approximately 20% have concomitant psychiatric conditions.

In line with disease severity, the CDC has more recently acknowledged that many COVID-19 survivors report symptoms that remain after the infection has disappeared [17]. Referred to as post-acute COVID-19 syndrome, it is considered present when symptoms associated with SARS-CoV-2 infection extend beyond 4 weeks. Post-acute COVID-19 sequelae appears relatively common. A study conducted by the CDC concluded that 65.9% of their study population that tested positive for SARS-CoV-2 reported symptoms that lasted greater than 4 weeks, with 76.2% reporting at least one persisting symptom [18]. Researchers described these symptoms as having cardiovascular, pulmonary, neurological, and psychological manifestations [19]. While the etiology of post-acute COVID-19 sequelae is unclear, autopsies performed on 44 patients with confirmed COVID-19 diagnosis reported measurable levels of SARS-CoV-2 RNA distributed throughout the body for as long as 230 days after the initial symptom onset [20]. However, other factors may also play a role. For example, although COVID-19 severity is greater in patients with certain medical conditions, including cardiovascular disease [21], one recent study has found that SARS-CoV-2 infections can cause cardiovascular dysfunction in patients who were otherwise healthy before their infection [22]. The spectrum of individuals with post-acute COVID-19 sequelae is alarming, especially given the growing evidence that these symptoms can impair activities of daily living [23]. Nonetheless, although the importance of post-acute COVID-19 sequelae is increasingly being recognized, the associations of COVID-19 disease severity with residual symptoms and pre-existing medical conditions remain poorly understood.

One gap in the existing literature is that prior studies were performed in hospitalized patients, leaving relatively little work describing the prevalence and symptoms of post-acute COVID-19 sequelae, or its risk factors, among individuals with less severe symptomatology. A prior study performed within the Long Island WTC first responder cohort reported that infection risk was highest in younger individuals and those who were employed [24]. The objective of this study was to examine physical and psychiatric antecedents associated with COVID-19 illness severity and post-acute COVID-19 sequelae presence among individuals who endorsed any COVID-19 infection. To examine COVID-19 disease severity and post-acute COVID-19 sequelae for individuals who were not previously hospitalized, we investigated lingering COVID-19 symptoms in an established prospective study of World Trade Center (WTC) responders [25] that has been monitoring a large cohort of WTC responders since 2011 [26]. Detailed health data were prospectively collected and provide a unique opportunity to study the effects of COVID-19 in this population. Despite being at low risk of death due to COVID-19, because of their relative youth, responders have multiple chronic health conditions including those affecting the respiratory and pulmonary systems, psychiatric conditions, PTSD, and depression that developed as a result of, or related to, toxic WTC exposures [27]. Many of the health conditions that are disproportionately prevalent among the WTC first responders have also been identified as potential risk factors for severe COVID-19 and death [8]. We hypothesized that a history of cardiovascular, gastrointestinal, and psychological diseases would be associated with a greater risk for disease severity as well as the development of post-acute COVID-19 sequelae.

## 2. Materials and Methods

### 2.1. Study Design and Participants

To meet the physical and mental health needs of first responders after the 9/11 attacks, the World Trade Center Health Program (WTCHP) was created [28]. As of 31 December 2021, the program had an enrollment of more than 82,000 responders nationwide and provided care to people who participated in the rescue and recovery efforts on 9/11 and the months thereafter [26]. At the Stony Brook University WTCHP, we prospectively followed approximately 13,000 responders for WTC-related and other conditions such as COVID-19 infection.

Study participants included members of the WTCHP cohort who were infected with SARS-CoV-2 before January 2022. To identify participants eligible for this study, clinic patients were screened with two COVD-19 questionnaires (Appendix A). Patients who answered “yes” to indicate a positive test result for COVID-19 between March 2020 and January 2022 through PCR testing, antigenic testing, or antibody testing were included in the COVID-19 cohort (N = 1549). Most cases were collected through the monitoring program, either before or during regularly scheduled visits. Additionally, a very small number were discovered through reports from case management. Patients were asked to provide evidence of a positive COVID-19 viral or antibody test result through laboratory testing or were found to have positive COVID-19 antibody results through nucleocapsid protein antibody testing performed at ARUP laboratories (testing code 3002776), Salt Lake City, Utah. Patients who did not have a positive COVID-19 result through any of these methods, or who had a positive spike protein antibody test after vaccination and a negative nucleocapsid protein antibody test, were excluded. English and Spanish language speakers were included.

### 2.2. Data Collection

Clinical information on initial and residual COVID-19 symptoms was collected from March 2020 through 2 February 2022. A data freeze was placed at that date because this study is an open cohort. In our cohort, the first confirmed case was in March 2020, and the last was in January 2022. This information was collected over time through in-person questionnaires and surveys sent via text and email, internal medical records, follow-up calls, and external medical records obtained with the release of health information forms. All symptom information was self-reported by the patients or collected from medical records at the time of COVID-19 diagnosis through a symptom checklist. Interviews were also conducted by phone or in person in real time as the study progressed, to complete the case reports. A database was created to streamline all gathered information into one easily accessible source. The database was separated into checklists based on clinical information, testing information, and report status. The symptoms checklist recorded the date of initial symptom onset. The documented symptoms reflected CDC guidelines during the time of data collection as well as several additional symptoms commonly reported by patients during their COVID-19 illness course [29]. Symptoms of COVID-19 in this study included systemic symptoms of fever, fatigue, headache, chills, and weight loss; respiratory symptoms of shortness of breath, chest pain, sore throat, congestion/runny nose, and wheezing; central nervous system symptoms of dizziness, vertigo, loss of smell/taste, and brain fog; psychiatric symptoms of anxiety, depression, and post-traumatic stress disorder (PTSD); gastrointestinal symptoms of nausea, vomiting, and diarrhea; and musculoskeletal symptoms of joint or muscle pain.

### 2.3. Measurements

#### 2.3.1. Demographic Data

Demographic variables of age, gender, and race/ethnicity were collected during registration and participants’ first clinical visits and were updated in follow-up visits as necessary. The analysis used the most recent update of demographic information before 1 January 2020 [28].

#### 2.3.2. COVID-19 Report Status: Complete and Incomplete

We categorized the study participants into *complete* or *incomplete* case status according to the verification of documented evidence of prior SARS-CoV-2 infection. Complete cases (N = 742) included patients for whom we were able to verify a history of a positive PCR, antigenic, or antibody test for SARS-CoV-2 or positivity for antibodies to SARS-CoV-2 nucleocapsid protein according to medical or laboratory records. Incomplete cases (N = 538) included patients who reported having a positive laboratory test for SARS-CoV-2, but for whom we were unable to access the documented proof of the test results.

#### 2.3.3. COVID-19 Severity: Asymptomatic, Mild, Moderate, Severe

Participants in the study were categorized into four groups according to their symptoms: asymptomatic (N = 129), mild (N = 511), moderate (N = 536), and severe (N = 104). This categorization was based on the NIH COVID-19 clinical spectrum updated October 2021 [30]. Appendix A provides details of the criteria used for categorization. Cases in the asymptomatic category were in patients who reported a positive SARS-CoV-2 virologic test result but did not report having any symptoms associated with COVID-19. The mild category included patients who had at least one symptom associated with COVID-19 but did not experience shortness of breath or difficulty breathing. These patients were medically managed mostly at home, even if they did initially visit a healthcare facility for medical treatment and/or testing. Moderate cases were in patients who reported shortness of breath and/or diagnosis of lower respiratory disease (pneumonia/bronchitis) during clinical assessment or imaging. These patients maintained SpO2 ≥ 94% on room air at sea level. Severe cases included patients with proven or endorsed SpO2 ≤ 93% on room air, and/or respiratory rate > 30 breaths/min, and/or heart rate greater than 100 beats per minute, and/or acute respiratory distress syndrome, septic shock, cardiac dysfunction, or an exaggerated inflammatory response in addition to pulmonary disease and/or severe illness causing cardiac, hepatic, renal, central nervous system, or thrombotic disease during COVID-19 illness. Patients were also categorized as severe if they otherwise endorsed hospital admission, supplemental oxygen use, ICU admission, or if death was from COVID-19.

#### 2.3.4. Post-Acute COVID-19 Symptoms

Residual symptoms were defined as any COVID-19-related symptoms that lasted at least 4 weeks after symptom onset [17]. Table 1 lists the definitions of four sub-symptoms of post-acute COVID-19 sequelae. The categories of residual symptoms were examined independently with each acute COVID-19 severity categorization.

#### 2.3.5. WTC-Related Diagnoses

Patients identified by a health care provider as having illnesses associated with 9/11 events can be certified for those illnesses and receive treatment [27]. The certifications are coded into categories including adjustment disorder (ADJ), anxiety (ANX), depression (DEP), PTSD, substance abuse (SA), extremity (EXT), head trauma (HT), spine (SP), gastroesophageal reflux disorder (GERD), interstitial lung disease (ILD), obstructive airway disease (OAD), sarcoid (SRC), and upper respiratory disease (URD) [31]. In this study, we analyzed the relationship between COVID-19 severity and post-acute COVID-19 sequelae and the following pre-existing comorbidities: URD, GERD, OAD, and depression. To measure depressive symptoms we used the Patient Health Questionnaire-9 (PHQ9) as a measure of depression [32]. In an effort to separate URD and OAD symptoms that can be similar such as cough and dyspnea, these certifications are determined based on systemic examination of the patients including repeated pulmonary function tests, physical examination, and medical history by a physician or nurse practitioner. After this examination, the documentation goes to the National Institute for Occupational Safety and Health (NIOSH) where it is further reviewed and categorized for approval of the certification [31].

#### 2.3.6. Other Health Indicators

Beyond pre-existing comorbidities, we analyzed a range of health indicators, including comorbidities other than certifications, that were hypothesized to be associated with COVID-19 severity. These indicators are usually measured during the annual health monitoring visits. The most recent pre-COVID-19 value was used in the analysis. We analyzed the following indicators of health: obesity (BMI ≥ 30), blood cholesterol level, hypertension, heart disease, and diabetes.

### 2.4. Statistical Analysis

Differences between groups were analyzed using the Student *t*-test or Kruskal–Wallis rank-sum test for continuous variables and the chi-square test or Fisher exact test for categorical variables. Significance of differences was analyzed using false detection rate (FDR)-adjusted *p*-value to adjust for multiple comparisons [33]. 

### 2.5. Severity

A multivariable model for the relationship between COVID-19 severity and potential risk factors was analyzed using ordinal logistic regression. This model assumes proportional odds meaning that the covariate coefficients are constant across different cut-offs of the ordinal dependent variable, but the intercept of each category is different [34]. Potential multicollinearity was diagnosed using the variance inflation factor (VIF) [35] in the R package “car” [36] (Appendix A). We provided the diagnostics of the proportional odds assumption in the Appendix A. In the multivariable model, age and PHQ9 were standardized by subtracting the mean value and dividing by the standard deviation to better compare the regression coefficients across the covariates. All analyses were performed in the R statistical computing environment [37]. The ordinal logistic regression analysis was implemented using the “MASS” package [38].

### 2.6. Post-Acute COVID-19 Syndrome

We included five outcome variables measuring post-acute COVID-19 sequelae: general post-acute COVID-19 sequelae, indicating whether participants had any post-acute COVID-19 sequelae; and post-acute COVID-19 respiratory sequelae, post-acute COVID-19 fatigue sequelae, post-acute COVID-19 CNS sequelae, and post-acute COVID-19 muscular sequelae, which were sub-elements of post-acute COVID-19 sequelae indicating whether participants had any post-acute COVID-19 sequelae in the respiratory system, fatigue, central nervous system, or muscular system, respectively.

To obtain reliable risk ratio estimates [39], we used Poisson regression with a binary outcome. The Poisson regression was implemented in R [37] via the “stats” package *glm* function. We first performed hypothesis testing with chi-square tests for categorical variables and Kruskal–Wallis rank-sum tests for each potential correlate. Then, we included all variables in one multivariable model. All *p*-values were FDR adjusted, which is the expected probability of false positives given all hypothesis tests conducted [33]. In the multivariable Poisson model, robust standard errors were used to estimate the coefficients and *p*-value via the “sandwich” package [40,41] to account for slight violation of the Poisson distribution assumption that the variance equals the mean [42]. 

## 3. Results

Among 1549 cohort members, 32 patients did not consent for medical information to be used in research, 1 had a negative test result at the time of data analysis, 197 had incomplete co-morbidity data, and 39 could not be categorized on the disease severity scale owing to incomplete medical information, thus leaving a final analytic sample size of 1280 patients.

### 3.1. Severity

Table 2 shows the characteristics of study participants in the entire study sample and separately among complete and incomplete cases. The two sub-samples appear to be similar with respect to all characteristics except COVID-19 severity. Complete cases have a higher proportion of severe cases, which is likely related to the fact that most severe cases had been hospitalized, including at the Stony Brook University Hospital, which made it easier to obtain access to their medical records related to the hospitalization episode and verify their testing results.

Table 3 shows the bivariate relationships between the covariates and COVID-19 severity. COVID-19 severity was significantly associated with age, gender, URD certification, OAD certification, GERD certification, obesity, diabetes, and depressive symptoms. COVID-19 severity increased with age, and males had a higher proportion of severe cases than females. At the same time, females had a higher proportion of moderate COVID-19 and a lower proportion of mild infections compared with males. Presence of 9/11 certifications (URD, OAD, and GERD) was associated with more severe COVID-19. Study participants having either of these certifications had higher proportions of moderate and severe COVID-19 compared with those who did not have the certifications. The depressive symptoms score, which measures the severity of depression, was significantly and substantially higher among people who had moderate or severe COVID-19 compared with asymptomatic and mild infections.

Table 4 shows the results of the multivariable ordinal logistic regression model for the association between COVID-19 severity (dependent variable) and the covariates in the univariate analysis (Table 2). After adjustment for covariates, diabetes and obesity were no longer statistically significant risk factors for severity. COVID-19 severity was significantly associated with age, black race, OAD certification, and depressive symptoms. The rest of the associations that were significant in the univariate analysis were not maintained in the multivariable regression.

Because of the nature of the dependent variable, we used an ordinal logistic regression to perform a multivariable analysis of the correlates of COVID-19 severity, measured on a four-category scale from mildest to most severe. The ordinal logistic regression involved an assumption of proportional odds, meaning that the odds ratio was assumed to be constant across all possible cut-offs. We tested the potential violation of this assumption and found that, overall, this assumption was likely to be violated in our analysis (Appendix A). Simultaneously, none of the significant covariates identified in the multivariable analysis violated the proportional odds assumption individually (Appendix A). To explore the sensitivity of our results to the violation of the proportional odds assumption, we conducted additional regression analyses by using binary logistic regression with three separate outcomes using different severity cut-offs: (1) asymptomatic vs. mild/moderate/severe; (2) asymptomatic/mild vs. moderate/severe, and (3) asymptomatic/mild/moderate vs. severe (Appendix A). The results of this analysis were generally consistent with the findings in the ordinal logistic regression analysis. 

### 3.2. Post-Acute COVID-19 Syndrome

#### Bivariable Associations

Table 5 shows the bivariable associations for the potential correlates and the main post-acute COVID-19 sequelae outcome. Acute COVID-19 severity, URD, GERD, OAD, heart disease, and depressive symptoms were significantly correlated with the presence of post-acute COVID-19 sequelae. Appendix A shows the same associations with four symptom categories specific to post-acute COVID-19 sequelae outcomes. Except for the consistent significant association with acute COVID-19 severity, URD, OAD, GERD, and depressive symptom were significantly associated with post-acute COVID-19 respiratory sequelae; GERD was significantly associated with post-acute COVID-19 CNS sequelae; and URD, GERD, and depressive symptoms were significantly associated with post-acute COVID-19 fatigue sequelae.

### 3.3. Multivariable Poisson Regression Models of Post-Acute COVID-19 Sequelae

Table 6 shows the Poisson regression results for the main post-acute COVID-19 sequelae outcome. Except for acute COVID-19 severity, only heart disease was significantly associated with the presence of post-acute COVID-19 sequelae. Subcategories were associated with COVID-19 severity as shown in the Appendix A; only the fatigue subcategory was associated with the presence of URD and depression severity.

## 4. Discussion

The current study investigated prospective associations between pre-existing risk factors and COVID-19 severity and post-acute COVID-19 sequelae in a large sample of responders to the 9/11 disaster who had COVID-19. Unlike many studies that have investigated the severity of COVID-19 by assessing potential risk factors retrospectively [10,12], the cohort analysis herein is nested within the open prospective cohort of WTC responders. This design provides many advantages, most notably the availability of accurate measurements of health and comorbidity profiles before SARS-CoV-2 infection. The study population, while unique in their exposures to the WTC disaster, did not experience worse outcomes due to their response efforts, rather, the outcomes were a result of the chronic health conditions they developed since. Therefore, results from this study can be generalized to the unexposed population with similar chronic health conditions. The results indicated that COVID-19 severity was independently associated with older age, Black race, OAD certification, and depression severity. In turn, COVID-19 disease severity was the strongest and the only factor significantly and consistently associated with the main post-acute COVID-19 sequelae outcome, as well as symptom-specific categories of post-acute COVID-19 sequelae. Taken together, the results contribute new evidence that both pre-existing respiratory and mental health conditions constitute risk factors for more severe COVID-19 symptoms, which, in turn, can put patients at a higher risk for long-term health sequela.

### 4.1. Severity

In our analysis, we identified several demographic risk factors that have been previously reported. For example, COVID-19 severity was significantly correlated with older age, as has been previously reported in a wide range of studies [6,8,9,10,12,14]. In addition, we found Black race was found to be associated with COVID-19 severity in those who have been infected after adjusting for other risk factors that have an association. This finding aligns with literature that has found significant increased COVID-19 disease severity in the African American population [7]. The CDC also recognizes that mental health conditions such as anxiety and depression increase the risk of severe COVID-19 [14]. A study on mental health has indicated that depression diagnosed later in life is associated with poorer outcomes of SARS-CoV-2 infection [43]. Similarly, we observed that more severe depressive symptoms, indicating greater depression severity, were associated with more severe COVID-19. 

Critical for this population, several WTC-related conditions previously reported to increase COVID-19 severity including OAD and URD were associated with more severe COVID-19. However, we also found that the relationship of URD became non-significant in multivariable analysis. Our findings support the literature indicating that chronic lung diseases may increase the likelihood of severe COVID-19 [12]. 

Some previously reported risk factors for severe COVID-19 were not confirmed in our analysis for severity, most notably hypertension, diabetes, and heart disease [10,26,27]. Individual analyses of conditions associated with cardiovascular health—obesity, hypertension, diabetes, heart disease, and GERD—may identify one or several, but not others, as risk factors for severity. Therefore, disentangling the primary cause from the mediating variables is difficult. Severe COVID-19 is on the causal pathway from many pre-existing conditions and post-acute COVID-19 sequelae and therefore would absorb the variation in the outcome making potential risk factors such as age or URD non-significant in the adjusted analysis.

### 4.2. Post-Acute COVID-19 Sequelae

Bivariate analysis of post-acute COVID-19 sequelae did initially find significant association with disease severity, certifications for URD, OAD, and GERD, depressive symptoms, and heart disease. Bivariate analysis of the respiratory, CNS, fatigue, and muscular subcategories did also initially find an association with severity. Additionally, all subcategories except muscular were associated with GERD. Respiratory and fatigue were associated with URD and depressive symptoms but only respiratory was associated with OAD.

Despite these findings, only severity and heart disease remained significant after multivariable analysis. In the literature, cardiovascular disease is recognized as a manifestation of post-acute COVID-19 sequelae [44]. Therefore, it is not a surprise that our analysis has shown an association of heart disease to post-acute COVID-19 sequelae. All subcategories maintained statistical significance in predicting severe COVID-19, but only fatigue showed association with URD and depressive symptoms. Consistent with prior studies, our findings suggest an association of post-acute COVID-19 sequelae association to URD [23]. URD, which is common (prevalence = 50%) in the WTC cohort, was associated with post-acute COVID-19 fatigue sequelae after adjustment for depressive symptoms and acute COVID-19 severity. Prior work has found an association of depression and depressive symptoms with post-acute COVID-19 sequelae [45], which we supported with our findings.

### 4.3. Limitations

Our study has several limitations. First, the study enrollment approach used to assemble the cohort for this analysis was likely to oversample moderate and severe infections, with respect to the COVID-19 severity distribution in the target population. A shift in the sample severity distribution with respect to population severity distribution was not expected to induce bias in the associations between COVID-19 severity and the indicators of health. However, the prevalence of severe COVID-19 reported in this sample should not be regarded as an estimate of this epidemiologic quantity in the population of WTC responders.

Although we made every possible effort to obtain documented proof of prior SARS-CoV-2 infection from the entire study population, some study participants were included based on self-reported infection. However, we note that the study participants for whom we were unable to obtain documented proof did report testing positive for SARS-CoV-2 through one of the available testing methods. Their self-reported prior infection was not based on the history of symptoms or suspected infection. The finding that the subsamples of complete and incomplete cases did not differ with respect to all characteristics except COVID-19 severity provides reassurance that misclassification was likely to be absent or minimal. We included a binary indicator of complete COVID-19 report status in all analyses to control for potential confounding. Authors should discuss the results and how they can be interpreted from the perspective of previous studies and of the working hypotheses. The findings and their implications should be discussed in the broadest context possible. Future research directions may also be highlighted.

## 5. Conclusions

In a cohort of WTC first responders, COVID-19 severity was significantly associated with age, Black race, OAD, and depression severity. From this, we found that increased disease severity was associated with the development of post-acute COVID-19 sequelae. WTC responders have a high prevalence of OAD and depression, due to their exposures at the World Trade Center disaster. Those responders who suffer from these conditions may be at elevated risk of severe COVID-19 and to post-acute COVID-19 sequelae. The implications of this study may be that the 9/11 disaster has increased the vulnerability of those who helped respond to the disaster and, therefore, that these exposures may have long-term implications that continue to affect the responders in a variety of ways many years after the attack. More broadly, our study confirmed several previously identified risk factors for severe COVID-19 that persist in various populations, thus indicating that greater precaution is warranted among responders and those with chronic airway diseases and mental health conditions.

## Figures and Tables

**Table 1 ijerph-19-06963-t001:** Post-acute COVID-19 Symptom Definitions.

Subcategory	Description
Respiratory	e.g., dyspnea, chest discomfort, cough, rhinitis, rhinorrhea, wheeze, sinusitis
Fatigue	e.g., “tired,” “low energy”
CNS	e.g., loss or reduction of smell/taste, mental fog, dizziness, vertigo, tinnitus, headache, balance issues
Musculoskeletal	e.g., myalgias, joint pain

**Table 2 ijerph-19-06963-t002:** Characteristics of the study participants (N = 1280) broken down by the complete/incomplete case report status.

	Total (N = 1280)	COVID-19 Report Status			^a^ FDR-p
		Completed (N = 742)	Incomplete (N = 538)		
Age				*t*-test	0.084
Mean (SD)	56.9 (7.37)	57.3 (7.38)	56.3 (7.31)		
Gender				Chi-square	0.931
Male	1170 (91.4%)	680 (91.6%)	490 (91.1%)		
Female	110 (8.6%)	62 (8.4%)	48 (8.9%)		
Race				Fisher exact	0.970
White	1096 (87.3%)	635 (87.1%)	461 (87.6%)		
Black	70 (5.6%)	42 (5.8%)	28 (5.3%)		
Hispanic	66 (5.3%)	40 (5.5%)	26 (4.9%)		
Other	23 (1.8%)	12 (1.6%)	11 (2.1%)		
Acute COVID Severity				Chi-square	<0.001
Asymptomatic	129 (10.1%)	78 (10.5%)	51 (9.5%)		
Mild	511 (39.9%)	289 (38.9%)	222 (41.3%)		
Moderate	536 (41.9%)	288 (38.8%)	248 (46.1%)		
Severe	104 (8.1%)	87 (11.7%)	17 (3.2%)		
Upper Respiratory Disease				Chi-square	0.378
Yes	778 (60.8%)	464 (62.6%)	314 (58.4%)		
No	501 (39.2%)	277 (37.4%)	224 (41.6%)		
Obstructive Airway Disease				Chi-square	0.378
Yes	450 (35.2%)	273 (36.8%)	177 (32.9%)		
No	829 (64.8%)	468 (63.2%)	361 (67.1%)		
Gastroesophageal Reflux Disorder				Chi-square	0.879
Yes	627 (49.0%)	368 (49.7%)	259 (48.1%)		
No	652 (51.0%)	373 (50.3%)	279 (51.9%)		
Obesity				Chi-square	0.088
Yes	707 (55.2%)	430 (58.0%)	277 (51.5%)		
No	573 (44.8%)	312 (42.0%)	261 (48.5%)		
Hypertension				Chi-square	0.088
Yes	420 (32.8%)	263 (35.4%)	157 (29.2%)		
No	859 (67.2%)	479 (64.6%)	380 (70.8%)		
Diabetes				Chi-square	0.879
Yes	130 (10.2%)	78 (10.5%)	52 (9.7%)		
No	1148 (89.8%)	663 (89.5%)	485 (90.3%)		
Heart Disease				Chi-square	0.433
Yes	131 (10.3%)	83 (11.2%)	48 (9.0%)		
No	1143 (89.7%)	655 (88.8%)	488 (91.0%)		
High Cholesterol				Chi-square	0.609
Yes	512 (40.1%)	305 (41.2%)	207 (38.6%)		
No	765 (59.9%)	436 (58.8%)	329 (61.4%)		
Depressive Symptoms				*t*-test	0.609
Mean (SD)	3.42 (4.47)	3.32 (4.28)	3.55 (4.73)		

**Note:** COVID-19: coronavirus disease 2019; post-acute COVID-19 sequelae: post-acute COVID-19 syndrome; SD: standard deviation; FDR-p: *p*-value after adjusting for the false discovery rate; ^a^ FDR adjusted *p*-value is the expected probability of false positives given all hypothesis tests conducted; *p*-value for the chi-squared or Fisher exact test for categorical variables and *t* test for continuous variables.

**Table 3 ijerph-19-06963-t003:** COVID-19 severity distribution among the study participants broken down by demographic characteristics and health variables.

	COVID-19 Severity					^a^ FDR-p
	Asymptomatic (N = 129)	Mild (N = 511)	Moderate (N = 536)	Severe (N = 104)	Method	
Age					Kruskal–Wallis	0.003
Mean (SD)	56.7 (8.51)	56.4 (6.89)	56.8 (7.15)	59.8 (8.53)		
Gender					Chi-square	0.024
Male	119 (10.2%)	476 (40.7%)	475 (40.6%)	100 (8.5%)		
Female	10 (9.1%)	35 (31.8%)	61 (55.5%)	4 (3.6%)		
Race					Fisher Exact	0.113
White	108 (9.9%)	455 (41.5%)	450 (41.1%)	83 (7.6%)		
Black	7 (10.0%)	19 (27.1%)	32 (45.7%)	12 (17.1%)		
Hispanic	7 (10.6%)	24 (36.4%)	27 (40.9%)	8 (12.1%)		
Other	2 (8.7%)	7 (30.4%)	13 (56.5%)	1 (4.3%)		
Upper Respiratory Disease					Chi-Square	<0.001
Yes	66 (8.5%)	281 (36.1%)	360 (46.3%)	71 (9.1%)		
No	63 (12.6%)	229 (45.7%)	176 (35.1%)	33 (6.6%)		
Obstructive Airway Disease					Chi-Square	<0.001
Yes	34 (7.6%)	131 (29.1%)	224 (49.8%)	61 (13.6%)		
No	95 (11.5%)	379 (45.7%)	312 (37.6%)	43 (5.2%)		
Gastro- esophageal Reflux Disorder					Chi-Square	<0.001
Yes	48 (7.7%)	218 (34.8%)	300 (47.8%)	61 (9.7%)		
No	81 (12.4%)	292 (44.8%)	236 (36.2%)	43 (6.6%)		
Obesity					Chi-Square	0.006
Yes	58 (8.2%)	279 (39.5%)	298 (42.1%)	72 (10.2%)		
No	71 (12.4%)	232 (40.5%)	238 (41.5%)	32 (5.6%)		
Hypertension					Chi-Square	0.048
Yes	37 (8.8%)	163 (38.8%)	173 (41.2%)	47 (11.2%)		
No	92 (10.7%)	347 (40.4%)	363 (42.3%)	57 (6.6%)		
Diabetes					Chi-Square	0.022
Yes	18 (13.8%)	40 (30.8%)	54 (41.5%)	18 (13.8%)		
No	111 (9.7%)	470 (40.9%)	481 (41.9%)	86 (7.5%)		
Heart Disease					Chi-Square	0.297
Yes	9 (6.9%)	49 (37.4%)	58 (44.3%)	15 (11.5%)		
No	118 (10.3%)	460 (40.2%)	477 (41.7%)	88 (7.7%)		
High Cholesterol					Chi-Square	0.415
Yes	43 (8.4%)	208 (40.6%)	217 (42.4%)	44 (8.6%)		
No	86 (11.2%)	300 (39.2%)	319 (41.7%)	60 (7.8%)		
Depressive Symptoms					Kruskal–Wallis	<0.001
Mean (SD)	2.54 (3.88)	2.70 (3.59)	4.06 (5.02)	4.72 (5.24)		

**Note:** COVID-19: coronavirus disease 2019; post-acute COVID-19 sequelae: post-acute COVID-19 syndrome; SD: standard deviation; FDR-p: *p*-value after adjusting for the false discovery rate; ^a^ FDR adjusted *p*-value is the expected probability of false positives given all hypothesis tests conducted; *p*-value for the chi-squared for categorical variables and Kruskal–Wallis test for continuous variables.

**Table 4 ijerph-19-06963-t004:** Correlates of COVID-19 severity using ordinal logistic regression model.

	OR	95% CI	FDR-p
Age ^a^	1.21	(1.06, 1.38)	0.015
Female	1.15	(0.78, 1.70)	0.583
Race: Black	2.01	(1.24, 3.27)	0.015
Race: Hispanic	1.22	(0.76, 1.95)	0.583
Race: Other	1.55	(0.72, 3.30)	0.408
Gastroesophageal Reflux Disorder	1.27	(1.00, 1.60)	0.131
Obstructive Airway Disease	1.86	(1.46, 2.38)	<0.001
Upper Respiratory Disease	1.16	(0.91, 1.48)	0.396
Obesity	1.16	(0.94, 1.45)	0.332
Hypertension	1	(0.78, 1.29)	0.999
High Cholesterol	0.94	(0.74, 1.18)	0.615
Heart Disease	1.13	(0.78, 1.62)	0.583
Diabetes	1.15	(0.80, 1.67)	0.583
Depressive Symptoms ^a^	1.27	(1.12, 1.43)	<0.001

**Note:** COVID-19: coronavirus disease 2019; post-acute COVID-19 sequelae: post-acute COVID-19 syndrome; aOR: multivariable-adjusted odds ratio; 95% CI: 95% confidence interval; FDR-p: *p*-value after adjusting for the false discovery rate; ^a^ continuous variables: age, and depressive symptoms were standardized by subtracting the mean and dividing by the standard deviation in this analysis. The odds ratios show associations between the dependent variable and one standard deviation increase in the respective continuous independent variable.

**Table 5 ijerph-19-06963-t005:** Patient Demographics and Other Conditions Stratified by Post-Acute COVID-19 Sequelae Status.

	Total (N = 1280)	No Post-Acute COVID-19 Sequelae (N = 853)	Post-Acute COVID-19 Sequelae (N = 366)	Method	FDR-p
Age				*t*-test	0.969
Mean (SD)	56.9 (7.37)	56.9 (7.27)	56.9 (7.41)		
Gender				Chi-square	0.397
Male	1170 (91.4%)	786 (92.1%)	329 (89.9%)		
Female	110 (8.6%)	67 (7.9%)	37 (10.1%)		
Race/Ethnicity				Chi-square	0.667
White	1096 (87.3%)	737 (87.9%)	306 (86.0%)		
Black	70 (5.6%)	44 (5.3%)	23 (6.5%)		
Hispanic	66 (5.3%)	42 (5.0%)	23 (6.5%)		
Other	23 (1.8%)	15 (1.8%)	4 (1.1%)		
Acute COVID Severity				Chi-square	<0.001
Asymptomatic	129 (10.1%)	118 (13.8%)	3 (0.8%)		
Mild	511 (39.9%)	391 (45.8%)	98 (26.8%)		
Moderate	536 (41.9%)	307 (36.0%)	206 (56.3%)		
Severe	104 (8.1%)	37 (4.3%)	59 (16.1%)		
Upper Respiratory Disease				Chi-square	<0.001
Yes	778 (60.8%)	486 (57.0%)	254 (69.4%)		
No	501 (39.2%)	366 (43.0%)	112 (30.6%)		
Obstructive Airway Disease				Chi-square	<0.001
Yes	450 (35.2%)	267 (31.3%)	160 (43.7%)		
No	829 (64.8%)	585 (68.7%)	206 (56.3%)		
Gastroesophageal Reflux Disorder				Chi-square	<0.001
Yes	627 (49.0%)	379 (44.5%)	220 (60.1%)		
No	652 (51.0%)	473 (55.5%)	146 (39.9%)		
Obesity				Chi-square	0.372
Yes	707 (55.2%)	463 (54.3%)	214 (58.5%)		
No	573 (44.8%)	390 (45.7%)	152 (41.5%)		
Hypertension				Chi-square	0.889
Yes	420 (32.8%)	282 (33.1%)	125 (34.2%)		
No	859 (67.2%)	570 (66.9%)	241 (65.8%)		
Diabetes				Chi-square	0.277
Yes	130 (10.2%)	81 (9.5%)	46 (12.6%)		
No	1148 (89.8%)	771 (90.5%)	319 (87.4%)		
Heart Disease				Chi-square	0.020
Yes	131 (10.3%)	74 (8.7%)	51 (14.0%)		
No	1143 (89.7%)	774 (91.3%)	314 (86.0%)		
High Cholesterol				Chi-square	0.635
Yes	512 (40.1%)	350 (41.2%)	141 (38.5%)		
No	765 (59.9%)	500 (58.8%)	225 (61.5%)		
Depressive Symptoms				Kruskal–Willis	<0.001
Mean (SD)	3.42 (4.47)	3.06 (4.23)	4.23 (4.71)		

**Note:** COVID-19: coronavirus disease 2019; post-acute COVID-19 sequelae: post-Acute COVID-19 syndrome; SD: standard deviation; FDR-p: *p*-value after adjusting for the false discovery rate.

**Table 6 ijerph-19-06963-t006:** Multivariable-adjusted risk ratios and 95% confidence intervals predicting the presence of Post-Acute COVID-19 Syndrome in World Trade Center responders.

	aRR	95% CI	FDR-p
Severity Asymptomatic	0.13	(0.04, 0.41)	0.002
Severity Moderate	1.82	(1.47, 2.26)	<0.001
Severity Severe	2.87	(2.23, 3.71)	<0.001
COVID-19 Report Status: Incomplete	1.13	(0.95, 1.34)	0.327
Age ^a^	0.98	(0.89, 1.08)	0.853
Female	1.12	(0.84, 1.49)	0.615
Race: Black	0.99	(0.70, 1.40)	0.971
Race: Hispanic	1.10	(0.81, 1.50)	0.708
Race: Other	0.71	(0.31, 1.64)	0.615
Gastroesophageal Reflux Disorder	1.22	(1.01, 1.48)	0.128
Obstructive Airway Disease	1.02	(0.85, 1.22)	0.954
Upper Respiratory Disease	1.19	(0.97, 1.46)	0.259
Obesity	1.00	(0.84, 1.19)	0.971
Hypertension	0.98	(0.81, 1.19)	0.954
High Cholesterol	0.91	(0.75, 1.09)	0.490
Heart Disease	1.34	(1.07, 1.67)	0.037
Diabetes	1.2	(0.95, 1.53)	0.295
Depressive Symptoms	1.07	(0.99, 1.16)	0.259

**Note:** COVID-19: coronavirus disease 2019; post-acute COVID-19 sequelae: post-acute COVID-19 syndrome; aRR: multivariable-adjusted risk ratio; 95% CI: 95% confidence interval; FDR-p: *p*-value after adjusting for the false discovery rate; ^a^ continuous variables: age and depressive symptoms were standardized by subtracting the mean and dividing by the standard deviation in this analysis. Odds ratios reported associations between the dependent variable and one standard deviation increase in the respective continuous independent variable.

## Data Availability

Medical information is protected, so only processed de-identified data will be made available upon receipt of a written request to the corresponding author.

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
