# Peer review of "The Impact of World Trade Center Related Medical Conditions on the Severity of COVID-19 Disease and Its Long-Term Sequelae"

_ijerph, 2022, doi:10.3390/ijerph19126963_

Round 1

Reviewer 1 Report

Article: The Impact of World Trade Center Related Medical Conditions on the Severity of COVID-19 Disease and its Long-Term Sequalae

Potential Conflicts of Interest: None

Review Report

            This study aimed to determine whether World Trade Center Related Illnesses may augment the risk for severe COVID-19 disease and post-acute COVID-19 syndrome in those participating in the rescue and recovery efforts associated with the 9/11 terrorist attacks on the World Trade Centers. Investigators prospectively followed a cohort of 1,280 individuals identified through the World Trade Center Health Program and through chart-review, surveys, and questionnaires elucidated the nature of the cohort’s COVID-19 symptoms, disease severity, and risk factors for severe disease and post-acute COVID syndrome. This work provides additional evidence that the responders to the 9/11 tragedy are a unique population unto themselves that require a special degree of clinical care, especially in light of the ongoing pandemic and its implications for medically complex special populations. The manuscript is well-written in terms of reporting, formatting, and overall organization. However, there are also several areas of the manuscript that require elaboration and/or clarification to determine the veracity of the results. Considerations are listed below.

General Concept Comments:

Article:

  1. Introduction – The introductory portion of this article provides a comprehensive overview of etiology of COVID-19, risk factors related to more severe outcomes of the disease, and post-acute COVID-19 syndrome, but given that the target journal is of the international persuasion, those outside of the U.S. may not be entirely familiar with the health-related consequences to those survivors of and first responders to the 9/11 terrorist attacks upon the World Trade Centers. It may be prudent to elaborate on exactly what these health-related issues are within the introduction of the article and why first responders and survivors face these consequences to help better frame the purpose of the study.
  2. Materials and Methods: Study Design and Participants – What exactly is the study design used here? Could be a bit more descriptive. Describe the questionnaire in brief to ensure that the approach to soliciting information from patients was unbiased as possible (e.g., no leading questions).
  3. Materials and Methods: Data Collection – It was mentioned that the study design was that of an open cohort. Were individuals disenrolled for any reason during the course of the study? It is stated that COVID-19 symptoms and sequelae were elicited from patients through surveys sent via text and email, what protocols were in place to ensure that the data obtained was provided by the intended recipient? If no safeguards were in place, this must be addressed in the limitations. Further, the limitations of self-reported outcomes and use of questionnaires and surveys to obtain information from a target population is not addressed at any point throughout the article. How was recall bias attended to in this study?
  4. Materials and Methods: COVID-19 Severity – It seems as though the participant categorization process based on the NIH COVID-19 clinical spectrum documentation was accomplished through some form of adjudication process. How many individuals were involved in this process? Were they blinded to one another’s selections?
  5. Materials and Methods: Other Health Indicators – Why weren’t the comorbidities of being overweight (i.e., 30 > BMI ≥ 25) and respiratory illnesses included?
  6. Materials and Methods: Statistical Analysis – Could be a bit more precise in terms of which specific variables are being analyzed via Student t-test/Kruskal-Wallis and Chi-square test/Fisher exact. Consider expanding upon why false-detection rates were employed.
  7. An argument could be made for both row and column % in many of your tables (such as Table 3).
  8. Discussion – Present limitations associated with patient-reported measures using questionnaires and surveys. Additional limitation related to being an uncontrolled analysis (there are no non-WTC responders included, correct?) and how this limits the insights possible here. Are there any differences between the risk factors of severity identified here versus the general population?

Specific Comments:

  1. Please present the WTCHP initialism after World Trade Center Health Program within parentheses on line 104, e.g., World Trade Center Health Program (WTCHP). Introduce the initialism first.
  2. On line 107, Long Island is mentioned, but what is significant about this geographic location and why is it important to the reader? Foreign readers may not be aware that this where Stony Brook University is. May be helpful to change to Stony Brook University.
  3. Lines 115 to 116 state that a small number of individuals were identified through case management. How many exactly were identified and how reliable is this approach in identifying these individuals?
  4. Lines 433-436: Authors did not delete the guiding text from the template.
  5. Recommend dropping “PACS” as an abbreviation and spelling out throughout the manuscript for the readers’ benefit.
  6. Abstract conclusion of: “We conclude that increased COVID-19 illness severity and the presence of PACS may be more common in WTC responders with chronic diseases resulting from exposures at the World Trade Center disaster.” ---- May need qualification relative to “compared to other WTC responders without chronic diseases” as it could read as “relative to non-WTC responders.

Rating the Manuscript:

Novelty: This study seems entirely novel; unable to find published scientific literature similar in scope that aims to elucidate the impact of COVID-19 in those with World Trade Center Related Medical Conditions.

Scope: This study fits within the scope of International Journal of Environmental Research and Public Health as it addresses topics related to occupational safety and health, infectious disease epidemiology, and health behavior, chronic disease, and health promotion. 

Significance: Results seem to be interpreted appropriately, significance is noted accordingly.

Quality: The article seems to be written in an appropriate manner with all results and analyses reported in an acceptable format.

Scientific Soundness: Generally speaking, the article is sound in its methodological approaches. There are some questions that this reviewer feels should be addressed to enhance the rigor of the study and to ensure the veracity of the results.

Interest to the Readers: Article is appropriate for target audience but is mostly of importance to those in the United States of America rather than a more international readership.

Overall Merit: This study does indeed build upon the existing understanding of the various risk factors that may predispose an individual to more severe COVID-19 illness all the while furthering the notion that first responders to the 9/11 disaster are a unique population unto themselves that require specialized care. I do believe that there is merit to reporting and publishing this study.

English Level: Proficient; article is easy to understand, and the syntax and structure are appropriate 

Overall Recommendation:

  1. Accept after Minor Revisions

Reviewer 2 Report

The choice of this peculiar cohort may have influenced the results, which nonetheless confirm what was previously known about the factors influencing both COVID-19 severity and occurrence of PACS. The presentation of the results should be improved.

Reviewer 3 Report

In a prior study performed within the Long Island WTC first responder cohort reported that infection risk for COVID-19 was highest in younger individuals and those who were employed.  The objective of this study was to examine physical and psychiatric antecedents associated with COVID-19 illness severity and post-acute COVID19 syndrome (PACS) presence among individuals who endorsed any COVID-19 infection. To examine COVID-19 disease severity and PACS for individuals who were not previously hospitalized, the authors investigated lingering COVID-19 symptoms in a previously established prospective study of World Trade Center (WTC) responders that has been monitoring a large cohort of WTC responders since 2011. Detailed health data were prospectively collected to study the effects of COVID-19 in this population. 

In the present study the investigators sought to determine whether certain conditions were risk factors for increased COVID-19 disease severity within a cohort of N=1,280 WTC responders with complete information on health outcomes prior to and following COVID-19 infection. They collected data on responders diagnosed with COVID-19, or had evidence of receiving positive SARS-CoV-2 polymerase chain reaction or antigen testing, or were asymptomatic but had IgG positive antibody testing. The presence of post-acute COVID-19 sequalae was measured using self-reported symptom severity scales.

Their results indicated that COVID-19 severity was independently associated with older age, Black race, obstructive airway disease certification, and depression severity. In turn, COVID-19 disease severity was the strongest and the only factor significantly and consistently associated with the main PACS outcome, as well as symptom-specific categories of PACS. Taken together, the results contribute new evidence that both pre-existing respiratory and mental health conditions constitute risk factors for more severe COVID-19 symptoms, which in turn can put patients at a higher risk for long-term health sequela.

General comments:

The paper is well-written with appropriate introduction, hypothesis, methods, results, discussion, limitations and conclusion sections. The English  is well-constructed and needs just  a little polishing.

Specific comments:

1. Page 4, Methods and Table 1:  Many symptoms of URD and OAD overlap (dyspnea, cough, wheeze, etc.). The authors should define how they separated these 2 conditions from each other when they conducted their bivariate and multivariate analyses. How were they able to distinguish between the two conditions?

2. P. 3, Data collection, ll. 1-3-123: While the cohort consisted of WTC first responders, it would be useful to know what their occupations were. By breaking down occupations, specific symptoms such as dyspnea, cough, wheeze can be linked to prior occupational exposure – such as with firefighters, paramedics, police officers, etc.  Similarly, occupations are often linked to depressive symptoms and PTSD.

3. P. 5, ll. 193-202:  Again, would be important to distinguish between URD and OAD – were criteria other than self-reported symptoms used, such as imaging, pulmonary function tests or imaging?

4. P. 6, Table 2:  Interesting demographics; I notice that males constituted the majority of the cohort – I assume this reflects the occupations of the first responders, e.g., firefighters, police, paramedics.  Also, Black responders are indicated as constituting only 5-6%, while Whites constituted 90% of the cohort.  Yet, government census statistics for 2022 indicate Black numbers in NYC are 24.3%, Whites only 42.7%. Please explain. I would assume even in 2011, NYC would be in the forefront of occupational diversity. Did many of the responders come from less diverse locations?  It is well-recognized that young individuals from rural areas have significantly less prevalence of airway reactivity than city-dwellers (presumably because they are desensitized to naturally found antigens at an early age). A review of the geographic origin of the responders might offer an explanation to this interesting finding. 

5. P. 8, ll. 271-280: The authors first state that COVID-19 severity was associated with a number of risk factors including obesity and diabetes (l. 273). Then further down, they say the same thing on line 280 – that obesity and diabetes were associated with more severe COVID-19. Were these 2 conclusions arrived at by different means, or is it just a repetition of the same finding? If so, the second statement should be deleted.

6. Table 4: Indicates that the FDR-p for Blacks was 0.015 – yet this population comprised only 5% of the entire cohort. Would the FDR still be significant?

7. P. 14, ll. 386-393:  Authors state that previously risk factors such as diabetes and obesity were not confirmed in their analysis for severity. Yet on p. 8, ll. 271-283 and in Table 3, they state that these conditions were indeed associated with more severe COVID-19. Please clarify:  Does this have anything to do with bivariate vs multivariate analysis?

8. P. 14, section 4.2, ll. 407-408:  Authors refer to “chronic airway disease” showing association PACS to URD. Again, what is the difference between chronic airway disease, URD and OAD? These need to be defined clearly.

9. P. 15, conclusions, ll. 444: Authors state that the 9/11 disaster increased the vulnerability of responders and that there may be long-term implications on their future health. But as I stated earlier, specific occupations may also predispose to a higher risk for COVID-19 severity. Firefighters and EMT personnel are exposed to burning buildings and materials as part of their occupation and this alone can increase their risk for COVID-19 severity. In other words, exposure to the WTC debris may not have been the only factor that contributed to severity.  It would be useful to analyze risk factors based on information regarding specific occupations. See comment #4 above.
